# First-Principles Mechanistic Insights into the Hydrogen Evolution Reaction on Ni₂P Electrocatalyst in Alkaline Medium

**Russell W. Cross and Nelson Y. Dzade \***

School of Chemistry, Cardiff University, Main Building, Park Place, Cardiff CF10 3AT, UK; crossrw@cardiff.ac.uk
\* Correspondence: DzadeNY@cardiff.ac.uk

**Abstract:** Nickel phosphide ($Ni_2P$) is a promising material for the electrocatalytic generation of hydrogen from water. Here, we present a chemical picture of the fundamental mechanism of Volmer–Tafel steps in hydrogen evolution reaction (HER) activity under alkaline conditions at the (0001) and ($10\bar{1}0$) surfaces of $Ni_2P$ using dispersion-corrected density functional theory calculations. Two terminations of each surface ($Ni_3P_2$- and $Ni_3P$-terminated (0001); and $Ni_2P$- and $NiP$-terminated ($10\bar{1}0$)), which have been shown to coexist in $Ni_2P$ samples depending on the experimental conditions, were studied. Water adsorption on the different terminations of the $Ni_2P$ (0001) and ($10\bar{1}0$) surfaces is shown to be exothermic (binding energy in the range of $0.33-0.68$ eV) and characterized by negligible charge transfer to/from the catalyst surface ($0.01-0.04$ $e^-$). High activation energy barriers ($0.86-1.53$ eV) were predicted for the dissociation of water on each termination of the $Ni_2P$ (0001) and ($10\bar{1}0$) surfaces, indicating sluggish kinetics for the initial Volmer step in the hydrogen evolution reaction over a $Ni_2P$ catalyst. Based on the predicted Gibbs free energy of hydrogen adsorption ($\Delta G_H$\*) at different surface sites, we found that the presence of $Ni_3$-hollow sites on the (0001) surface and bridge Ni-Ni sites on the ($10\bar{1}0$) surface bind the H atom too strongly. To achieve facile kinetics for both the Volmer and Heyrovsky–Tafel steps, modification of the surface structure and tuning of the electronic properties through transition metal doping is recommended as an important strategy.

**Keywords:** hydrogen evolution reaction (HER); water splitting; earth-abundant materials; nickel phosphides; density functional theory

## 1. Introduction

Rising global energy demands and the serious concerns of environmental contamination necessitate the development of renewable energy sources to alleviate our reliance on fossil fuels and simultaneously satisfy increasingly stringent environmental regulations. Molecular hydrogen ($H_2$) is considered a promising energy carrier to meet future global energy demands owing to its high energy density and environmentally benign characteristics [1,2]. Among the several sources of $H_2$ generation, the electrocatalytic hydrogen evolution reaction (HER) from water splitting is the most economical and effective route for a future hydrogen economy [3,4]. Although Pt-group metals are currently considered the best HER electrocatalysts owing to the optimum Gibb's free energy of hydrogen adsorption ($\Delta G_{H*} \approx$ 0.09 eV) [5–7], the high cost and scarcity of these noble metal-based electrocatalysts severely limits their scalable applications [8]. Therefore, the development of cost-effective and earth-abundant catalysts as possible alternatives has received significant research interest recently.

Non-noble metal-based electrocatalysts—such as tungsten and molybdenum chalcogenides [9], carbides [10], nitrides [11], phosphides [12,13], and phosphosulfides [14,15] have received considerable attention. Among the reported catalysts, transition metal phosphides such as CoP [16], FeP [17],

and Ni$_2$P [18–20] have been regarded as very promising electrocatalysts for the HER owing to their low-cost, appropriate electronic structures, and high electrochemical stability. Nickel phosphide (Ni$_2$P) in particular is an emerging catalyst for hydrogen evolution reactions [20–23], hydrodesulfurization (HDS) [24,25], hydrodenitrogenation (HDN) [26–28], hydrodeoxygenation (HDO) [29], hydrodechlorination (HDCl) [30], and water–gas shift reactions [31]. Ni$_2$P is considered an attractive alternative to noble metal catalysts for HER [20] because both of its constituent elements nickel and phosphorous are cheap, abundant, and non-toxic, which makes Ni$_2$P a promising cost-effective material for scalable renewable energy conversion systems.

Nickel phosphides are stable and durable in strong acid and alkali conditions, prolonging the turnover number (TON) and lifetime of the catalyst and hence can achieve enhanced HER efficiency [32,33]. The Ni$_2$P (0001) surface is the most studied facet for HER activity owing to its comparable predicted hydrogen evolution activity to that of hydrogenase [22,34–36]. Earlier investigations have considered HER activity in acidic medium over Ni$_2$P catalysts, whereby the Volmer–Tafel mechanism (H$^+$$_{(aq)}$ → H$^*$, 2H$^*$ → ↑H$_2$) involves only characterizing the Gibb's free energy hydrogen adsorption to the catalyst surface [22,23]. There exist limited studies of the HER activity of Ni$_2$P under alkaline conditions and the underlying mechanism is still poorly understood [37,38]. The mechanism of the HER in alkaline media is slightly different to acidic media and can typically be treated as a combination of three elementary steps: the Volmer step—water dissociation and formation of a reactive intermediate (H$_2$O + e$^-$ + cat → H*-cat + OH$^-$)—followed by either the Heyrovsky step (H*-cat + H$_2$O + e- → cat + OH$^-$ + ↑H$_2$) or the Tafel recombination step (2H*-cat → ↑H$_2$). Thus, the HER activity of an electrocatalyst in alkaline conditions is dominated by the prior Volmer step and subsequent Tafel step (i.e., Volmer reaction is the rate-determining step for the HER in alkaline electrolytes) [39].

The HER activity is often reported to be severely hindered by sluggish kinetics and high overpotential over Ni$_2$P catalyst [12]. However, an atomic-level picture of the nature of the Ni$_2$P catalyst surfaces and active sites that dictate the fundamental adsorption of water and the subsequent Volmer–Tafel steps in the hydrogen evolution reaction over Ni$_2$P catalyst has not been investigated comprehensively. This information and insights are, however, vital in the quest to rationally design more active and stable Ni$_2$P electrocatalysts. In this study, first-principles density functional theory (DFT) calculations were employed to investigate the mechanisms of water adsorption and dissociation (H$_2$O → OH$^-$ + H$^+$) over Ni$_2$P. The investigation provides insight of thermodynamic stability, active sites of adsorption, and activation barriers on the Ni$_2$P (0001) and (10$\bar{1}$0) surfaces. The geometries for molecular water adsorption were first investigated, followed by climbing image nudged elastic band (CI-NEB) methodology to determine the activation energy barriers for HER on each surface. The Gibbs free energy of hydrogen adsorption $\Delta G_H{}^*$, which quantifies the strength of H adsorption was calculated to provide insight into the HER activity of the different surfaces and terminations of the Ni$_2$P electrocatalyst in an alkaline environment.

## 2. Results and Discussion

### 2.1. Ni$_2$P Bulk and Surface Characterization

Nickel phosphide (Ni$_2$P) crystallizes in the hexagonal structure (Figure 1a) and belongs to space group P$\bar{6}$2$m$ with a = b = 5.859 Å and c = 3.382 Å [40]. The fully optimized bulk Ni$_2$P cell parameters (a = b = 5.821 Å and c = 3.320 Å) are within 2% of experimental values [40]. The calculated Ni−Ni and Ni−P bond lengths are 2.589 Å and 2.289 Å, which are both also within 2% of the experimental values [41]. The calculated partial density of states (PDOS) shown in Figure 1b reveals the metallic character of Ni$_2$P, whereby the Ni $d$-states dominate the regions around the Fermi level, which is consistent with earlier DFT results [23,42].

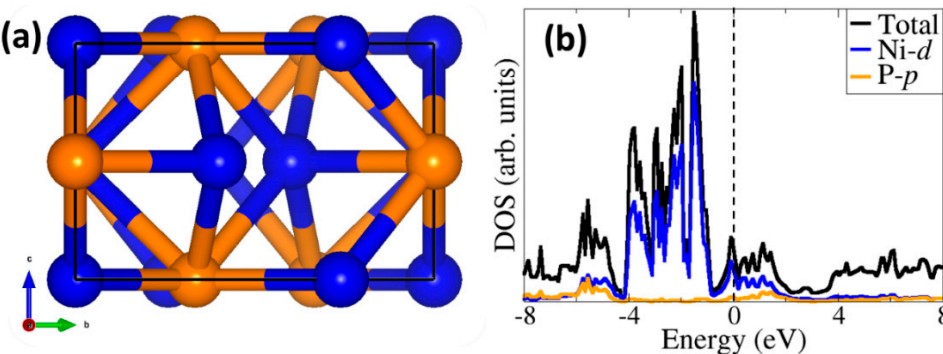

**Figure 1.** (**a**) Hexagonal structure of bulk $Ni_2P$ and (**b**) projected density of states (PDOS) showing the metallic character of $Ni_2P$. Colour code: Ni = blue, P = orange.

The $Ni_2P$ (0001) and (10$\bar{1}$0) surfaces were created from the fully relaxed bulk structure in order to eliminate any residual forces which may be present during surface relaxation. All possible terminations that ensure symmetric slab models were investigated in order to determine the most stable termination. The symmetric slab model helps to avoid unphysical overall polarization of the slab and ensures a zero-dipole moment in the z-direction, perpendicular to the surface plane. The (0001) and (10$\bar{1}$0) surfaces each have two symmetric slab terminations, both of which were investigated in order to determine the most stable termination. The (0001) can be $Ni_3P_2$- or $Ni_3P$-terminated (Figure 2a,b). The $Ni_3P_2$ termination of the (0001) surface has relatively equivalent nickel sites—each with two neighbouring phosphorous and conjugating within a 3-Ni triangle ($Ni_3$-hollow site), with average Ni−Ni bond length of 2.665 Å. At the $Ni_3P$-terminated (0001) surface, each phosphorous is coordinated to three nickel atoms. The $Ni_3P_2$- and $Ni_3P$-terminated (0001) surfaces possess similar surface energies calculated at 1.91 and 1.92 $Jm^{-2}$, respectively, which suggests that both terminations could be formed (coexist), depending on the experimental conditions. In an earlier theoretical investigation [35], where the formation energy as a function of the chemical potential of the P atom ($\mu_P$) for the $Ni_3P_2$-and $Ni_3P$-terminated surfaces was derived, it was shown that both $Ni_3P_2$-and $Ni_3P$-terminations have comparable range of thermodynamic stability, although over a wide range of $\mu_P$ ($-10.10$ eV $< \mu_P <$ $-6.62$ eV), there is preference for the $Ni_3P_2$ terminations. A scanning tunneling microscopy (STM) study of the (0001) surface validates the existence of both the $Ni_3P_2$- and $Ni_3P$-terminations of the (0001) surface [43]. Similar to the (0001) surface, the (10$\bar{1}$0) surface has two terminations: $Ni_2P$- and NiP-terminated surfaces as shown in Figure 2c,d—whereby the $Ni_2P$-termination has been observed experimentally, and the NiP is included due to the closeness in surface energies. The $Ni_2P$-termination of the (10$\bar{1}$0) surface shows there are two distinct nickel sites, differentiated by the number of neighbouring phosphorous atoms—having either one or two coordinated P atoms. The NiP-termination of the (10$\bar{1}$0) surface exposes a single nickel and phosphorous atom at the topmost layer (Figure 2d). Both terminations are also found to have similar surface energies, calculated at 1.33 and 1.39 $Jm^{-2}$ for the $Ni_2P$- and NiP-terminations, respectively, which again suggests that both terminations could be formed depending on the experimental conditions. The (10$\bar{1}$0) has been less extensively studied, and so the same amount of literature does not exist as for the (0001)-surface. Previous computational work has focused on the (10$\bar{1}$0)-$Ni_2P$ termination [22]. However, in 2010, K. Edamoto et al. [44] did confirm two ordered surfaces exist on the (10$\bar{1}$0) surface from their photoemission spectroscopy (PES) analysis. The predicted lower surface energy of the (10$\bar{1}$0) compared to the (0001) surface, however, indicate that the (10$\bar{1}$0) surface is thermodynamically more stable. The relative stabilities, geometries, and compositions of the $Ni_2P$ (0001) and (10$\bar{1}$0) surfaces may dictate their chemical reactivity toward water adsorption and splitting. This has been investigated and discussed in the subsequent sections.

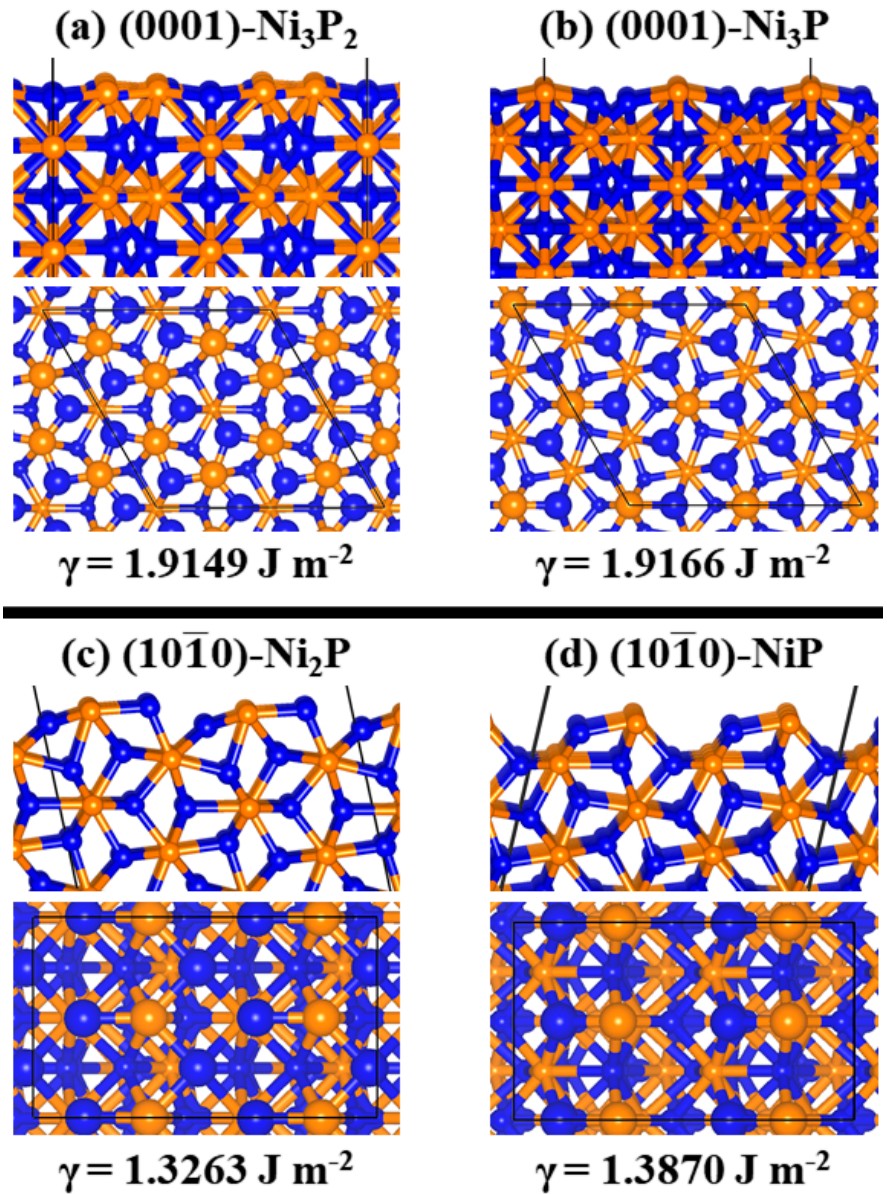

**Figure 2.** Schematic illustration of the side (top row) and top (bottom row) views of the relaxed structures of the (**a**) Ni$_3$P$_2$- and (**b**) Ni$_3$P-terminations of (0001) surface, (**c**) Ni$_2$P- and (**d**) NiP-terminations of (10$\bar{1}$0) surface. $\gamma$ denotes the surface energy.

*2.2. Molecular H$_2$O Adsorption on Ni$_2$P (0001) and (10$\bar{1}$0) Surfaces*

Seeing that the Volmer reaction (H$_2$O + e$^-$ + cat → H*-cat + OH$^-$) is the rate-determining step for the HER in alkaline electrolytes, the first interest is to characterize the adsorption and activation of water on the Ni$_2$P (0001) and (10$\bar{1}$0) surfaces. The most stable adsorption geometries for molecular water on the Ni$_3$P$_2$- and Ni$_3$P-terminated (0001) surface are shown in Figure 3, whereas the calculated adsorption energies and optimized structural parameters are listed in Table 1. The adsorption of water on the Ni$_3$P$_2$- and Ni$_3$P-terminated (0001) surface released adsorption energies of −0.55 and −0.59 eV, respectively. The similarity in the adsorption energies is consistent with the relative stability of the two terminations. The interacting Ni−O bond distance at the Ni$_3$P$_2$- and Ni$_3$P-terminated (0001) surfaces is calculated at 2.219 Å and 2.165 Å, respectively. The O−H bond lengths calculated for water on each surface (Table 1) were slightly larger than for a free H$_2$O molecule (0.972 Å) in vacuum. This indicates that the O–H bonds of the adsorbed species are slightly weakened when adsorbed on the Ni$_2$P (0001) surface—which was confirmed via O–H bond stretching vibrational frequencies (Table 1). Due to the

fact that vibrational modes and bond strengths may be related to charge transfer processes and electron density rearrangement between the surface species and the adsorbing water molecule (Figure 3c,d), Bader population and differential charge density analyses were performed to quantify any charge transfer and electron density redistribution within the $Ni_2P-H_2O$ systems. Overall, it was found that the adsorption of water on both the $Ni_3P_2$- and $Ni_3P$-terminated (0001) surface is characterized by only small charge transfer ($< 0.03$ e$^-$), suggesting a small degree of water oxidation (Table 1). The calculated electron density difference isosurface plots (Figure 3c,d) reveal accumulation of electron density within the bonding region of Ni−O, which is consistent with chemisorption. Evidence of electron density redistribution between hydrogen and surface species can be also seen, suggestive of hydrogen-bonded interactions that we believe contributes to the stabilization of adsorbed water on the catalyst surface.

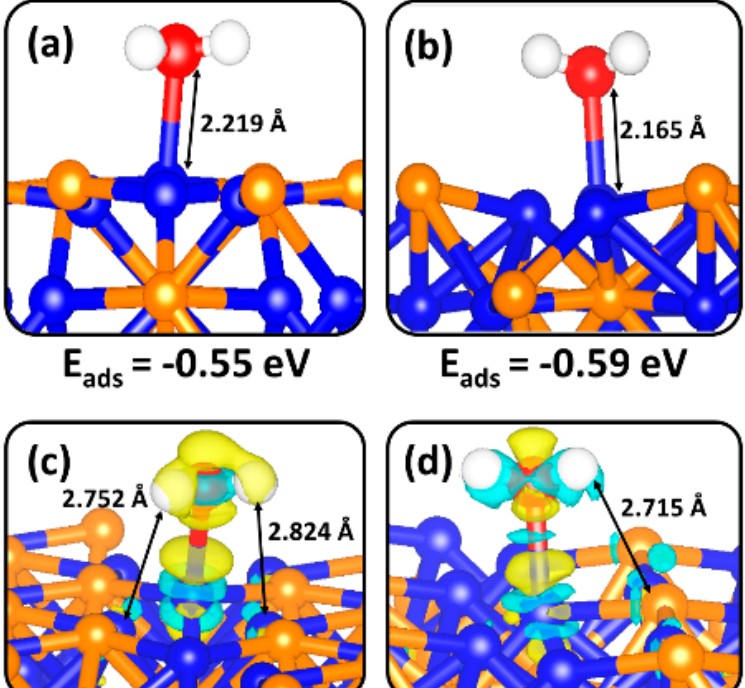

**Figure 3.** The optimized structures of the most favourable adsorption configurations of molecular $H_2O$ on the (**a**) $Ni_3P_2$- and (**b**) $Ni_3P$-terminations of (0001) surface. The corresponding isosurfaces contours of the charge density difference for adsorbed $H_2O$ molecule are shown in (**c**) and (**d**). The cyan and yellow contours indicate electron density accumulation and depletion by $\pm 0.003$ e/Å$^3$, respectively.

**Table 1.** Molecular adsorption energies ($E_{ads}$), bond lengths (d), bond angles ($\alpha$), Bader charge of adsorbed water ($q(H_2O)$), and vibrational frequencies ($\upsilon$) for water adsorption on $Ni_2P$ (0001) and (10$\bar{1}$0) surfaces. The gas phase O–H stretching modes are predicted at 3988 and 3866 cm$^{-1}$.

| Surface | $E_{ads}$ (eV) | $d$(Ni-O) (Å) | $d$(O-H) (Å) | $\alpha$(H-O-H) (°) | $q(H_2O)$ (e-) | $\upsilon$(O-H) (cm$^{-1}$) |
|---|---|---|---|---|---|---|
| (0001)-$Ni_3P_2$ | −0.55 | 2.219 | 0.980 | 104.6 | 0.04 | 3820, 3705 |
| (0001)-$Ni_3P$ | −0.59 | 2.165 | 0.976 | 106.8 | 0.02 | 3829, 3735 |
| - | - | - | - | - | - | - |
| (10$\bar{1}$0)-$Ni_2P$ | −0.68 | 2.153 | 0.988 | 101.8 | 0.01 | 3607, 3493 |
| (10$\bar{1}$0)-$Ni_2P$ | −0.33 | 2.298 | 0.984 | 105.0 | -0.01 | 3632, 3540 |
| (10$\bar{1}$0)-NiP | −0.58 | 2.156 | 0.995 | 104.5 | 0.03 | 3763, 3574 |

The lowest-energy molecular $H_2O$ geometries at the two terminations of the (10$\bar{1}$0) surface are shown in Figure 4. Two stable adsorption water geometries are obtained on the $Ni_2P$-terminated (10$\bar{1}$0) surface, which released adsorption energies of −0.68 eV when adsorbed at edge Ni-site (Figure 4a) and −0.33 eV when adsorbed at step Ni-site (Figure 4b). The stronger binding at the edge Ni-site

can be attributed to its lower coordination number (N = 4) compared to that at a step Ni-site (N =6). At the NiP-terminated ($10\bar{1}0$) surface, the lowest-energy water adsorption geometry (Figure 4c) released adsorption energy −0.58 eV. The interacting Ni–O bond distances at the $Ni_2P$-terminated ($10\bar{1}0$) surface are calculated at 2.152 Å and 2.298 Å for the lowest-energy structure (Figure 4a) and next stable structure (Figure 4b). At the NiP-terminated ($10\bar{1}0$) surface (Figure 4c), the Ni–O bond distance is 2.156 Å. We observe small elongation in the O–H bonds of the adsorbed water molecule compared to the gas phase molecule, indicating that the O–H bonds are somewhat activated. Bader charge analysis reveals only small charge transfer between the surface-Ni and O of the adsorbed water molecule (Table 1). The calculated differential charge density isosurface (Figure 4d–f)) reveals electron density depletion from the Ni sites and accumulation in the Ni–O bonding regions which is consistent with chemisorption. Evidence of hydrogen-bonded interrelations is also seen in the accumulated electron densities in regions between the hydrogen and the adjacent surface P-sites.

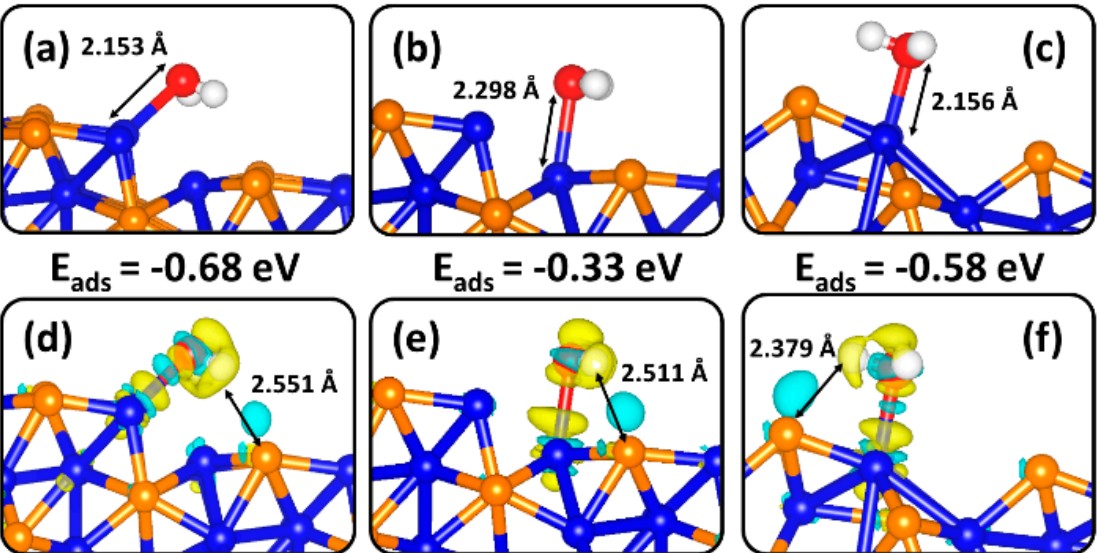

**Figure 4.** The most favourable, fully relaxed binding conformations of molecular $H_2O$ on the (**a,b**) $Ni_2P$- and (**c**) NiP-terminations of the ($10\bar{1}0$) surface. The corresponding isosurfaces contours of the difference in charge density for an adsorbed $H_2O$ molecule are shown in (**d–f**). The cyan and yellow contours indicate electron density accumulation and depletion by ±0.003 e/Å³, respectively.

### 2.3. Dissociative $H_2O$ Adsorption on $Ni_2P$ (0001) and ($10\bar{1}0$) Surfaces

To gain atomic-level insights into the fundamental mechanism of HER activity on $Ni_2P$ surfaces, the thermodynamics and kinetic energy barrier of water dissociation (i.e., the Volmer step) were systematically studied. The stretched O–H bonds observed for $H_2O$ adsorption suggest that these molecular adsorbed conformations are likely precursors for initiating the Volmer step of HER. $Ni_2P$ surface structures will dictate their water splitting activity, hence this is the current focus of our investigation. Different co-adsorption structures for OH and H species were considered on the $Ni_2P$ (0001) and ($10\bar{1}0$) surfaces in order to determine lowest-energy configurations. Shown in Figure 5i–l and Table 2 are the co-adsorption energies and structural parameters for the lowest-energy configurations. The structures and energetics of the less-stable explored co-adsorption geometries are shown in Supplementary Information Figures S1 and S2, and Table S1. The most preferred co-adsorption geometry on the (0001)-$Ni_3P_2$ termination is found for the configuration in which the OH species form bidentate Ni–O bonds with the H atom binding at the 3-fold Ni ($Ni_3$-hollow) site (Figure 5i), releasing a co-adsorption energy of −1.07 eV. Compared to the (0001)-$Ni_3P_2$-terminated surface, the adsorption energy for dissociative water adsorption has been found to be endothermic at all explored sites on the $Ni_3P$-terminated (0001) surface, with the least endothermic energetics (0.45 eV, Figure 5j) predicted for the geometry in which the OH binds at top-Ni site and the H atom bridges Ni-P site. At the ($10\bar{1}0$)-$Ni_2P$

terminated surface, the most favourable dissociative adsorption structure is found for the configuration in which both OH and H species preferentially bind at adjacent bridge Ni–Ni sites (Figure 5k) releasing a co-adsorption energy of −0.69 eV. Compared to the $Ni_2P$-terminated ($10\bar{1}0$) surface, unfavourable (endothermic) dissociative adsorption is obtained at all binding sites on the NiP-terminated ($10\bar{1}0$) surface with the least endothermic energy (0.05 eV) predicted for the configuration in which OH and H bind at adjacent top-Ni and top-P sites, respectively, as shown in Figure 5k.

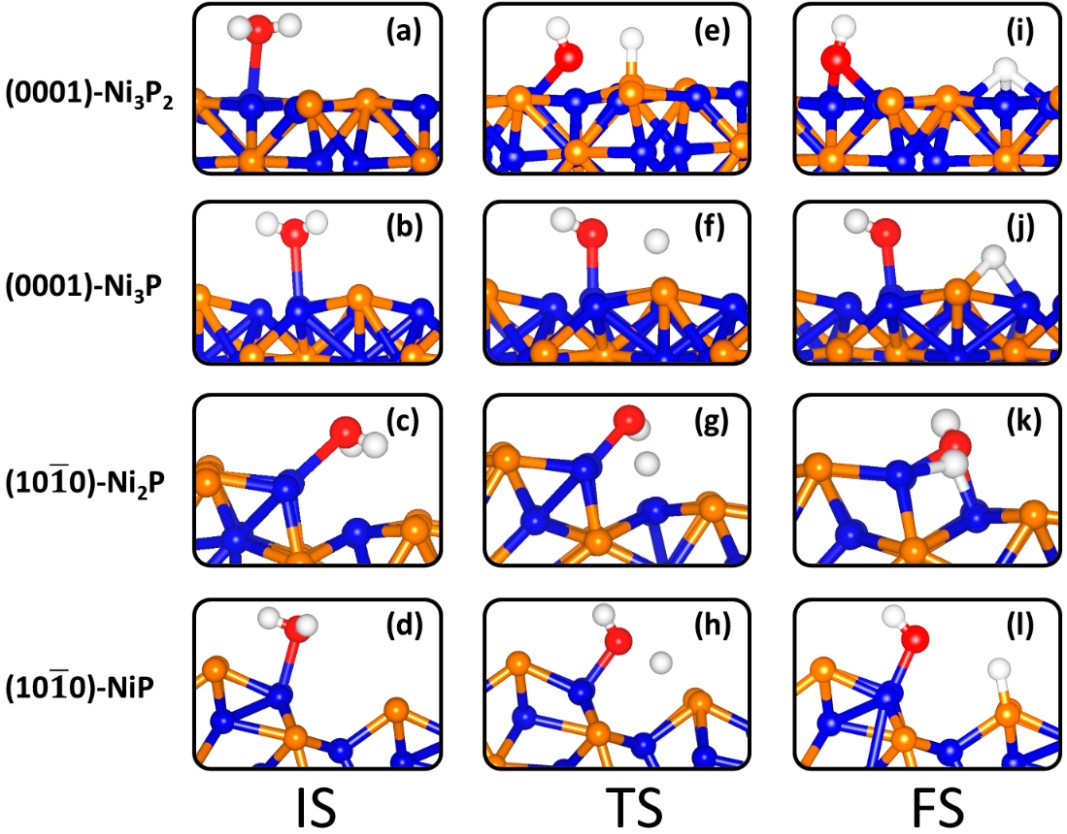

**Figure 5.** The initial (IS) (**a**–**d**), transition (TS) (**e**–**h**), and final (FS) (**i**–**l**) states of the most favourable path for water dissociation on the (0001) and ($10\bar{1}0$) surfaces of $Ni_2P$.

**Table 2.** Dissociative adsorption energies ($E_{ads}$) and bond lengths (d) for water on $Ni_2P$ (0001) and ($10\bar{1}0$) surfaces. H-P bond distances are denoted as d(P).

| Surface | Site (OH/H) | $E_{ads}$(OH + H) (eV) | d(Ni–O) (Å) | d(O–H) (Å) | d(H–Ni/(P)) (Å) |
|---|---|---|---|---|---|
| (0001)-$Ni_3P_2$ | bridge-Ni/$Ni_3$-hollow | −1.07 | 2.034 | 0.978 | 1.761 |
| (0001)-$Ni_3P$ | top-Ni/bridge-NiP | 0.45 | 1.860 | 0.977 | 1.746, 1.547 (P) |
| ($10\bar{1}0$)-$Ni_2P$ | bridge-Ni/bridge-Ni | −0.69 | 1.969 | 0.975 | 1.582 |
| ($10\bar{1}0$)- NiP | top-Ni/top-P | 0.05 | 1.841 | 0.980 | 1.427 (P) |

To ascertain whether the dissociation or desorption of water will take place on the $Ni_2P$ catalyst surfaces, we have calculated the activation energy barriers for water dissociation on the (0001) and ($10\bar{1}0$) surfaces (Figure 6) and compared it with the molecular water adsorption energies on the corresponding surfaces (Table 1). The schematic representations of the initial (IS), transition (TS), and final (FS) states of the dissociation of water on the $Ni_2P$(0001) and $Ni_2P$($10\bar{1}0$) surfaces are shown in Figure 5. The adsorption energy of the water molecule on each termination of the $Ni_2P$(0001) and $Ni_2P$($10\bar{1}0$) surfaces are obviously smaller, in absolute value, than the energy required for water dissociation (1.53 and 1.12 eV on the $Ni_3P_2$- and $Ni_3P$-terminated (0001) surfaces; 0.96 and 0.82 on the $Ni_2P$- and NiP-terminated ($10\bar{1}0$) surfaces, Figure 6), suggesting sluggish kinetics for the initial Volmer

step in the hydrogen evolution reaction over $Ni_2P$ catalyst, and thus requiring a high overpotential for the dissociation to happen [45].

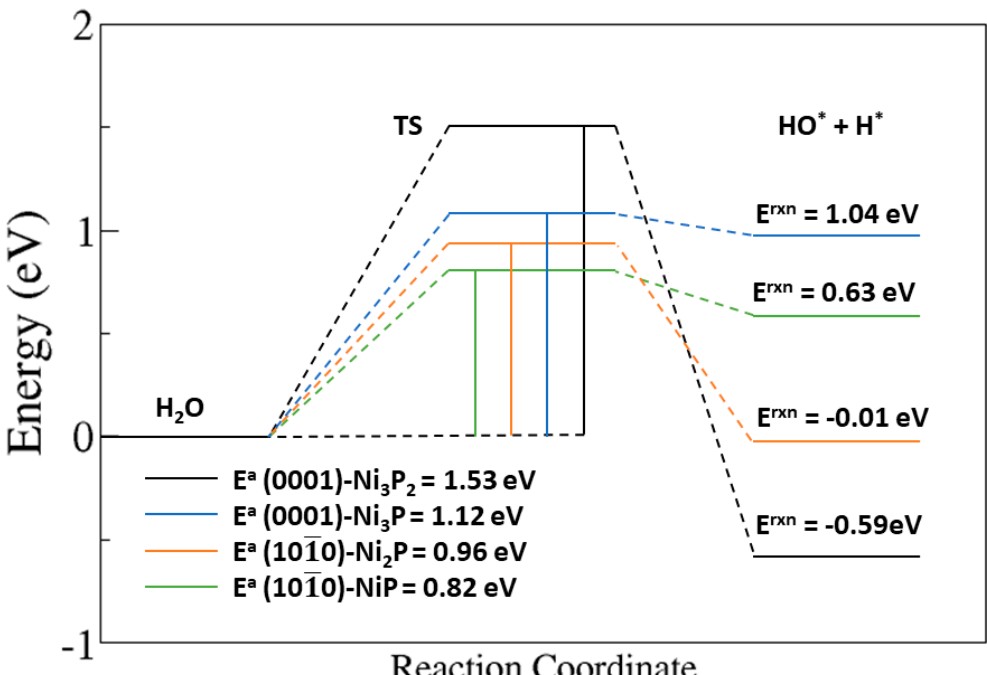

**Figure 6.** Activation barriers ($E^a$) and reaction energies ($E^{rxn}$) for dissociative water adsorption to the (0001) and ($10\bar{1}0$) surfaces of $Ni_2P$.

## 2.4. H Adsorption on $Ni_2P$ (0001) and ($10\bar{1}0$) Surfaces

After the splitting of water ($H_2O \rightarrow H + OH$) i.e., the Volmer step, the subsequent combination of the adsorbed H* into molecular hydrogen (Tafel step) is vital in HER. The Gibbs free energy of hydrogen adsorption ($\Delta G_H^*$), which quantifies the strength of hydrogen adsorption on the catalyst surfaces is known to be the best descriptor for hydrogen evolution activity [4,7]. For an active HER catalyst, the value of $|\Delta G_H^*|$ must be close to zero, indicating that the free energy of adsorbed H is close to that of the reactant or product. The $\Delta G_H^*$ value was calculated using the relation $\Delta G_{H*} = \Delta E_H + \Delta E_{ZPE} - T\Delta S_H$, where $\Delta E_H$ is the hydrogen adsorption energy calculated as $\Delta E_{H*} = E_{Ni_2P+H} - E_{Ni_2P} - \frac{1}{2}E_{H_2}$. $\Delta E_{ZPE}$ is the difference in zero-point energy between the adsorbed hydrogen and hydrogen in the gas phase, and $\Delta S_H$ is the entropy difference between adsorbed state and gas phase. The vibrational and configurational entropies of the adsorbed H*-intermediate are assumed to be negligible, and thus the entropy difference is simply $\Delta S_H \approx -\frac{1}{2}S_{H_2} = -0.7$ meV K$^{-1}$, where $S_{H_2}$ is the entropy of molecule hydrogen in gas phase. Under standard conditions, $\Delta E_{ZPE} - T\Delta S_H$ is about 0.24 eV, therefore $\Delta G_{H*} = E_H + 0.24 \; eV$ [4,7,46].

Different binding sites were explored for H adsorption on the $Ni_2P$ (0001) and ($10\bar{1}0$) surfaces in order to determine the optimum $\Delta G_H^*$ value. The optimized adsorption structures with the predicted $\Delta G_H^*$ values at the $Ni_2P$ (0001) and ($10\bar{1}0$) surfaces are shown in Figures 7 and 8, respectively. The calculated optimum $|\Delta G_H^*|$ values (Figure 9) at the $Ni_3P_2$- and $Ni_3P$-terminated (0001) surfaces were calculated at 0.13 eV and 0.20 eV, respectively, predicted at the top-Ni (Figure 7a) and bridge Ni-P (Figure 7e) sites. The small $|\Delta G_H^*|$ values indicate a facile Tafel step on both terminations, but the smaller $\Delta G_H^*$ value for the $Ni_3P_2$-termination (0.13 eV) suggest a superior HER activity. Compared to the top-Ni and bridge Ni-P sites on the $Ni_3P_2$-terminated (0001) surface, the $Ni_3$-hollow site binds H too strongly ($\Delta G_H^* = -0.42$ eV, Figure 7d) similar to the reported value of $-0.45$ eV by Wexler et al. [47]. The stronger binding of H at the $Ni_3$-hollow site points to sluggish Heyrovsky–Tafel step as the diffusion of the adsorbed H* to combine is restricted. At the $Ni_2P$- and NiP-terminated ($10\bar{1}0$) surfaces,

the optimum $|\Delta G_H*|$ value is calculated to be 0.34 eV (Figure 8a) and 0.01 eV (Figure 8d), suggesting the NiP-terminated (10$\bar{1}$0) surface provides a nearly thermoneutral H adsorption and consequently higher catalytic activity towards HER. Similar to the Ni$_3$-hollow site at the Ni$_3$P$_2$-terminated (0001) surface, we found that the Ni−Ni (2.797 Å) bridge site at the Ni$_2$P-terminated (10$\bar{1}$0) surface binds H too strongly ($\Delta G_H*$ = −0.44, Figure 8b).

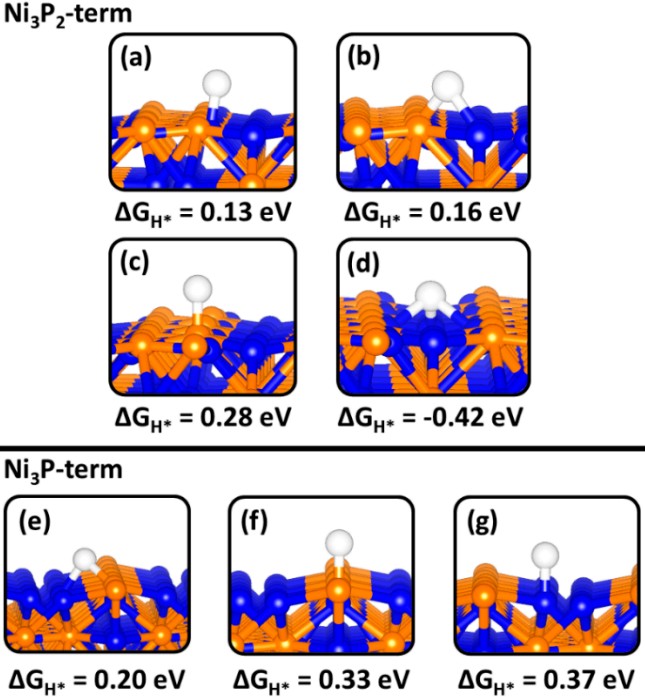

**Figure 7.** Gibbs free of hydrogen adsorption ($\Delta G_{H*}$) at different binding sites on the (0001) surface of Ni$_2$P. Whereby (**a**–**d**) represent Ni-top, Ni-P-bridge, P-top and 3-Ni-bridge, respectively on the Ni$_3$P$_2$-termination. Along with (**e**–**g**) showing a Ni-P-bridge, P-top and Ni-top respectively on the Ni$_3$P-termination.

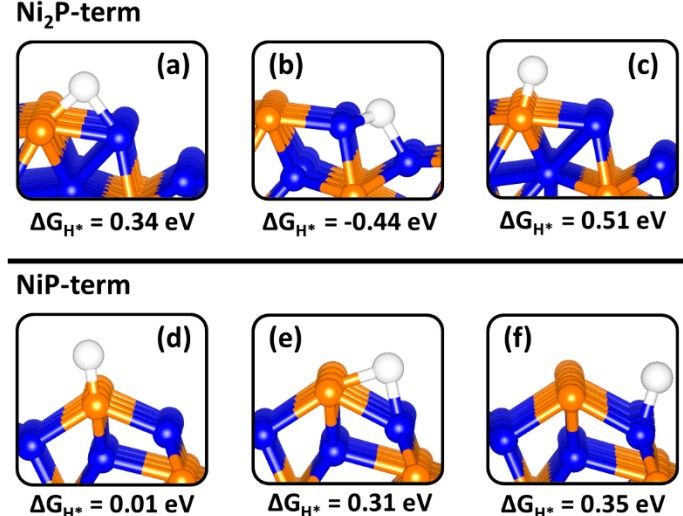

**Figure 8.** Gibbs free of hydrogen adsorption ($\Delta G_{H*}$) at different binding sites on the (10$\bar{1}$0) surface of Ni$_2$P. Whereby (**a**–**c**) show a Ni-P-bridge, Ni-Ni-bridge and P-top respectively on the (10$\bar{1}$0)-Ni$_2$P termination and (**d**–**f**) are the P-top, Ni-P-bridge and Ni-top respectively on the (10$\bar{1}$0)-NiP termination.

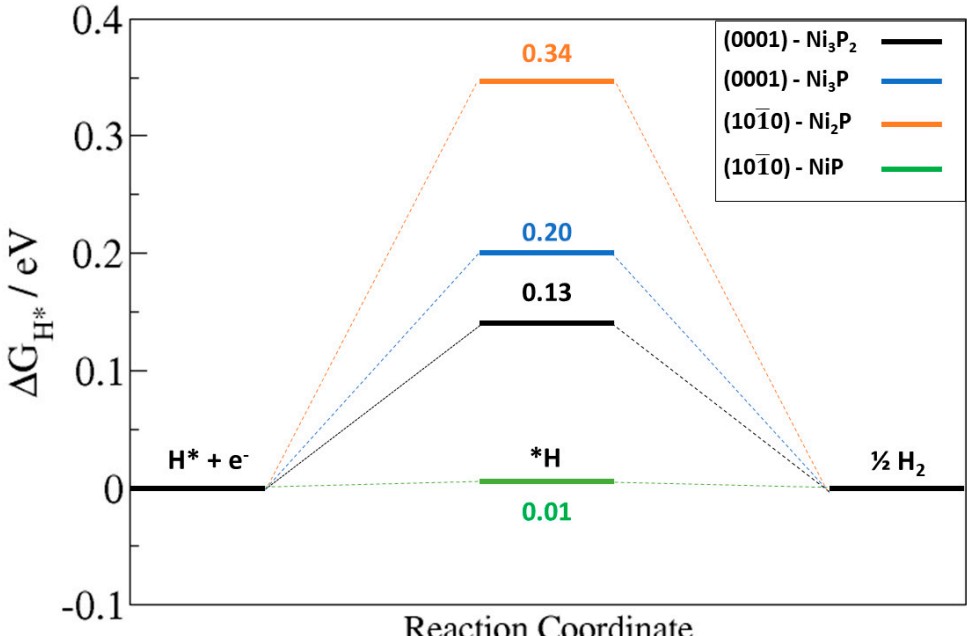

**Figure 9.** The optimum Gibbs free of hydrogen adsorption ($|\Delta G_{H*}|$) on (0001) and ($10\bar{1}0$) surfaces of $Ni_2P$.

## 3. Summary and Conclusions

The adsorption and dissociation reaction mechanisms of water over the $Ni_2P$ (0001) and ($10\bar{1}0$) surfaces have been investigated by means of first-principles DFT-D3 calculations. Results regarding the thermodynamic stability, active sites, and activation barriers of both the Volmer and Tafel steps have been systematically characterized. The dissociative adsorption of water is demonstrated to be thermodynamically favourable (exothermic) at both the $Ni_2P$ (0001) and ($10\bar{1}0$) surfaces, with activation energy barrier of 1.12 and 0.82 eV, respectively. The calculated dissociative adsorption energetics and high activation energy barriers for the dissociation, however, suggest sluggish kinetics for the initial Volmer step in the hydrogen evolution reaction over a $Ni_2P$ catalyst. This indicates that a high overpotential is required to drive the HER activity on pristine $Ni_2P$ catalyst. Based on predicted $\Delta G_H*$ values at different surface sites, we predict that the $Ni_3$-hollow sites on the (0001) and the Ni-Ni bridge sites on the ($10\bar{1}0$) surfaces bind H too strongly, hence the modification of the surface structure and tuning of the electronic properties through transition metal doping could be an important strategy to achieve facile kinetics for both the Volmer and Heyrovsky–Tafel steps. Future investigations will therefore expand the results presented here to detail transition metal doped $Ni_2P$ catalysts. This will provide full atomic-level insights into the geometric and electronic properties that dictate their improved HER activity. The insights derived from the present study on the pristine $Ni_2P$ (0001) and ($10\bar{1}0$) surfaces in terms of surface stability, active sites, adsorption energies and activation barriers for water dissociation will be useful in investigating the ligand (changes in the electronic structure) and ensemble (structure sensitivity) effects of transition metal doped $Ni_2P$ electrocatalysts towards achieving improved HER activity.

## 4. Computational Details

All the calculations were performed using the Vienna Ab Initio Simulation Package (VASP) [48]. The generalized gradient approximation (GGA) with the Perdew–Burke–Ernzerhof (PBE) was used for the calculation of the electronic exchange–correlation potential [49]. The projector augmented wave (PAW) method was used to describe the electron-ion interactions [50]. Wave functions were expanded in a plane wave basis with a high energy cut-off of 600 eV. The convergence criterion was set to $10^{-6}$ eV between two ionic steps for the self-consistency process, with Hellmann–Feynman forces on

each ion reached 0.01 eV Å$^{-1}$. We have accounted for Van der Waals dispersion forces through the Grimme DFT-D3 scheme [51], which adds a semi-empirical dispersion correction to the conventional Kohn–Sham DFT energy as implemented in the VASP code. The coefficients of the R$^{-6}$ term in the DFT-D3 scheme is influenced by the neighborhood of each atom and they are they continuously change along with environment changes during geometry optimization. The Brillouin zone was sampled using a Monkhorst-Pack [52] *k*-point grid of $5 \times 5 \times 11$ for the bulk and $3 \times 3 \times 1$ for the surface calculations.

The METADISE code [53,54] was employed to create the Ni$_2$P (0001) and (10$\bar{1}$0) surfaces (Figure 1), which are known to be active for HER [19,22] from the fully relaxed bulk material. The slab thickness of the (0001) and (10$\bar{1}$0) surfaces which ensured convergence of the surface energy within 0.001 eV is found to be 13.28 Å and 10.08 Å, respectively. A vacuum size of 15 Å was added in the z-direction of the slab to avoid interactions between periodic slabs. The surface energy ($\gamma$) of each surface termination was calculated using the relation

$$\gamma = \frac{E_{surface} - n\,E_{bulk}}{2\,A} \tag{1}$$

where $E_{\text{surface}}$ and $E_{\text{bulk}}$ are the energies of fully surface and bulk structures, respectively, *n* is the number of repeat unit cells in the z-direction, and A is the surface area.

A ($2 \times 2$) supercells of the Ni$_2$P (0001) and (10$\bar{1}$0) surfaces (Figure 2) was employed for the characterization of the adsorption structures and properties of water. The area of these surfaces is large enough to reduce or avoid lateral interactions between water and its dissociated products in periodic cells. To obtain the lowest-energy adsorption geometries, atoms of the topmost three surface layers and the adsorbate species were allowed to relax without constraints until the residual forces reached 0.01 eV Å$^{-1}$. The strength of water adsorption on the different Ni$_2$P surfaces was characterized by calculating the adsorption energy ($E_{ads}$) using the equation

$$E_{ads} = E_{surface+adsorbate} - \left( E_{surface} + E_{adsorbate} \right) \tag{2}$$

where $E_{surface + adsorbate}$ is the energy of the slab with adsorbed molecule, $E_{surface}$ is the energy of the naked surface, and $E_{adsorbate}$ is the energy of adsorbates in the gas phase. Based on this definition, a negative or positive value of $E_{\text{ads}}$ indicates an exothermic or endothermic adsorption process. Bader charge analysis [55,56] was used to quantify any charge transfers between the Ni$_2$P surface and adsorbate molecules. Transition state structures (confirmed via frequency calculations) and the corresponding activation energy barriers for water dissociation was determined using the climbing image nudged elastic band (cNEB) method [57].

**Supplementary Materials:** The following are available online at http://www.mdpi.com/2073-4344/10/3/307/s1, Figure S1: The optimized structures of dissociated water configurations (C) on the Ni$_3$P$_2$- and Ni$_3$P-terminations of Ni$_2$P(0001) surface; Figure S2: The optimized structures of dissociated water configurations (C) on the Ni2P- and NiP-terminations of the Ni2P (10$\bar{1}$0) surface, and Table S1: Dissociative adsorption energies (*E*ads) and bond lengths (*d*) for water on Ni2P (0001) and (10$\bar{1}$0) surfaces. H-P bond distances are denoted as *d*(P). Sites Ni, P, Ni-Ni and Ni-P denote top-Ni, top-P, bridge-Ni-Ni and bridge-Ni-P.

**Author Contributions:** R.W.C. performed the DFT simulations and data analysis and wrote the paper. N.Y.D. contributed to the study design and scientific discussion of the results. Both co-authors contributed to the manuscript. All authors have read and agreed to the published version of the manuscript.

**Funding:** This research was funded by the UK's Engineering and Physical Sciences Research Council (EPSRC), grant number EP/S001395/1.

**Acknowledgments:** R.W.C. acknowledges the College of Physical Sciences and Engineering, Cardiff University for studentship. We also acknowledge the use of computational facilities of the Advanced Research Computing at Cardiff (ARCCA) Division, Cardiff University, and HPC Wales. Information on the data that underpins the results presented here, including how to access them, can be found in the Cardiff University data catalogue at http://doi.org/10.17035/d.2020.0102983129.

**Conflicts of Interest:** The authors declare no conflict of interest.

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
