# Peer review of "First-Principles Mechanistic Insights into the Hydrogen Evolution Reaction on Ni2P Electrocatalyst in Alkaline Medium"

_catalysts, doi:10.3390/catal10030307_

Round 1

Reviewer 1 Report

In this paper nickel phosphide (Ni2P) has been theoretically investigated as electrocatalyst for the hydrogen evolution reaction. Two surfaces (0001 and 1010) have been analyzed via periodic density functional theory. The thermodynamics and kinetics of the Volmer and Heyrovsky-Tafel steps have been explored, showing that the water adsorption is thermodynamically favorable on both the considered surfaces. However, the high activation energy for the water dissociation, suggests sluggish kinetics for the Volmer step.

Overall, the scope of the paper is good, and the discussion on the results is convincing.

Moreover, the insights provided in this paper will be a good starting point for the future design of metal-doped Ni2P catalysts.

The paper deserves publication in Catalysts.

Reviewer 2 Report

The manuscript describes a study of hydrogen evolution reaction on Ni2P catalyst via first principles-based simulations. Possible adsorption sites of the reactants, their adsorption energies, and activation energies of the reaction are computed. It is concluded that dissociative adsorption of water molecules (known as the Volmer step) will be sluggish. Also, the Gibbs free energy of hydrogen adsorption is too high on some sites on the catalyst and that will make the recombination of hydrogen atoms less likely. Moderate modification of the catalyst may enhance the reaction kinetics in the future. Overall, the subject is very important and more studies are needed. This study has shown some progress on understanding hydrogen evolution reaction on Ni2P catalyst and I recommend publishing the work with some minor corrections.   Corrections/comments:   The title suggests "alkaline medium". I failed to understand why the medium is alkaline. Elaboration on this will help.   line 62-63: should it be OH-?   line 77: Gibbs free energy   Table 1 caption does not mention what 'q' is.   Provide isosurface value in the caption of Fig 3(c)-(d) and Fig 4 (d)-(f)   line 167: "distance is obtained at 2.156" -> distance is 2.156   line 212: possibly wrong phrase: 'terminations of the Ni 2 P(0001)'
line 233: A reference is needed for the value of "entropy of hydrogen molecule in the gas phase".
line 263: "The adsorption ..." - > "The dissociative adsorption"   line 269: should there be a modulus on ΔG?   line 283: not sure why the references 42 and 48 are cited there.   line 285-286: unclear statement "the convergence criterion was set to 10−6 eV between two ionic steps for the self-consistency process." There should be two criteria: i) for self-consistency of the wavefunction (within one ionic step) and ii) either energy convergence between two successive SCF steps or atomic force convergence. I see later on line 291 force criteria is mentioned.   line 288: this phrase is not entirely correct "..semi-empirical dispersion potential..." As far I know, the Grimmer D3 correction is not a functional of electron density. It adds only energy and does not contribute to the derivative of energy with respect to density (i.e. the DFT potential). An appropriate phrase is "...semi-empirical dispersion energy".  

Reviewer 3 Report

In this manuscript, authors provided fundamental mechanisms and comparison for hydrogen evolution reaction (HER) catalyzed by nickel phosphide. They compared Ni2P (0001) and (1010) surfaces well to conclude high over-potential for water dissociation and strong H binding sites. What they suggest is a transition metal doping to modify surface structure and electronic property. It would be perfect if they add any doping they mentioned for future work in this manuscript. A question is the following.

1. Authors confirmed the Ni3P2- and Ni3P-terminations of (0001) surface in Ref. [43]. What about (1010) surfaces? Any reference literature for Ni2P- and NiP-termination?
